# Phosphorus-Solubilizing Bacteria Enhance Cadmium Immobilization and Gene Expression in Wheat Roots to Reduce Cadmium Uptake

**DOI:** 10.3390/plants13141989

**Published:** 2024-07-21

**Authors:** Delong Kan, Minyu Tian, Ying Ruan, Hui Han

**Affiliations:** 1Key Laboratory of Hunan Provincial on Crop Epigenetic Regulation and Development, College of Bioscience and Biotechnology, Hunan Agricultural University, Changsha 410128, China; kandelongphd@163.com (D.K.); tianminyu1995@126.com (M.T.); 2Collaborative Innovation Center of Water Security for the Water Source Region of the Mid-Line of the South-to-North Diversion Project of Henan Province, Nanyang Normal University, Nanyang 473061, China

**Keywords:** phosphorus-solubilizing bacteria, wheat, cadmium, phenylalanine metabolism, transcriptome, cadmium fluorescence staining

## Abstract

The application of phosphorus-solubilizing bacteria is an effective method for increasing the available phosphorus content and inhibiting wheat uptake of heavy metals. However, further research is needed on the mechanism by which phosphorus-solubilizing bacteria inhibit cadmium (Cd) uptake in wheat roots and its impact on the expression of root-related genes. Here, the effects of strain *Klebsiella aerogenes* M2 on Cd absorption in wheat and the expression of root-related Cd detoxification and immobilization genes were determined. Compared with the control, strain M2 reduced (64.1–64.6%) Cd uptake by wheat roots. Cd fluorescence staining revealed that strain M2 blocked the entry of exogenous Cd into the root interior and enhanced the immobilization of Cd by cell walls. Forty-seven genes related to Cd detoxification, including genes encoding peroxidase, chalcone synthase, and naringenin 3-dioxygenase, were upregulated in the Cd+M2 treatment. Strain M2 enhanced the Cd resistance and detoxification activity of wheat roots through the regulation of flavonoid biosynthesis and antioxidant enzyme activity. Moreover, strain M2 regulated the expression of genes related to phenylalanine metabolism and the MAPK signaling pathway to enhance Cd immobilization in roots. These results provide a theoretical basis for the use of phosphorus-solubilizing bacteria to remediate Cd-contaminated fields and reduce Cd uptake in wheat.

## 1. Introduction

Cadmium (Cd), one of the most toxic heavy metals, is highly mobile and is widely present in water and soils [1]. Cd enters animals and humans through the food chain and poses a serious threat to human health [2]. In recent years, the continuous exposure of the “cadmium wheat” incident has caused widespread concern among all of society about the issue of excessive Cd in wheat [3,4]. Wheat plants mainly absorb Cd through their roots, where the element accumulates in the plant body, affecting cell structure, photosynthesis and enzyme activity and strongly inhibiting crop growth [5]. Moreover, Cd levels in wheat grains exceeding the national standard can have harmful effects on human health [6]. In addition, the scarcity of available phosphorus is an important problem. These soils are rich in phosphorus, but the phosphorus available for absorption and utilization by plants is insufficient [7]. Excessive use of fertilizers can lead to phosphorus fertilizer loss and water pollution. Therefore, approaches that both reduce Cd uptake by wheat and increase the available phosphorus content in the soil are urgently needed.

Phosphorus-solubilizing microorganisms secrete organic acids and acid phosphatases through their own metabolism, dissolving insoluble phosphates in the soil and converting unavailable phosphorus to forms available to plants [8]. In addition, phosphorus-solubilizing microorganisms reduce heavy metal bioavailability through cell wall adsorption and extracellular precipitation, providing technical means for the safe utilization of heavy metal-contaminated farmlands [9,10]. Khourchi et al. [11] reported that spraying the phosphorus-solubilizing bacterium *Bacillus siamensis* together with polyphosphates increased the amount of available phosphorus in the soil and promoted wheat growth. The combination of phosphorus-solubilizing bacteria and biochar promoted acid secretion and extracellular electron transfer and helped bacteria resist Cd and Pb toxicity [12]. In our previous research, a Cd-immobilizing and phosphorus-solubilizing bacterium, *Klebsiella aerogenes M2*, was isolated from heavy metal-contaminated vegetable fields [13]. Strain M2 released available phosphorus into the soil, thereby promoting radish (*Raphanus sativus* L.) growth and inhibiting Cd absorption by radish plants. However, the effects of strain M2 on Cd uptake by wheat and its underlying mechanism have not been thoroughly studied.

When plants are exposed to heavy metal stress, the genes in the roots also undergo certain changes. High-throughput RNA transcriptome sequencing can be used to identify key differentially expressed genes (DEGs) [14]. Many genes related to Cd detoxification or Cd absorption, including natural resistance-associated macrophage protein 5 (*Nramp5*), heavy metal ATPase 2 (*OsHMA2*), low-affinity cation transport protein 1, and natural resistance-related macrophage protein 1, have been identified in various plant species [15,16]. The key DEGs involved in resistance to Cd in the roots of wheat [17], rice (*Oryza sativa* L.) [18], maize (*Zea mays* L.) [15], and muskmelon (*Cucumis melo* L.) [19] have been screened by transcriptome technology. In addition, the molecular mechanism (gene level) by which the exogenous addition of silicon (Si) [20], manganese (Mn) [21], nanomaterials [22], or fungi [23] inhibits Cd uptake in wheat has been studied via transcriptomic methods. By regulating the antioxidant system, *Bacillus altitudinis* WR10 increases the resistance of wheat to Cu. The expression of genes related to phenylpropanoid biosynthesis significantly increased in wheat roots, which may improve phenolic acid accumulation to protect plant cells from Cu toxicity [24]. However, the molecular mechanism of Cd immobilization and Cd detoxification in wheat roots mediated by the phosphorus-solubilizing bacterium M2 has not been reported.

In this study, we investigated (1) the effects of strain M2 on Cd uptake by wheat roots and the underlying physiological and biochemical mechanisms, which were studied hydroponic experiments, and (2) the molecular mechanisms of Cd detoxification and immobilization in wheat roots mediated by strain M2, which were studied via transcriptomic technology. The research results provide a theoretical basis for microbial passivation and remediation of Cd-contaminated wheat fields.

## 2. Results

### 2.1. Mechanisms of Cd Immobilization by Strain M2

The results of the solution adsorption test revealed that strain M2 could grow well in inorganic phosphorus-containing medium, in which the source of phosphorus was water-insoluble calcium phosphate. On the first day of cultivation, the OD_600_ was 1.11, and on the third day of cultivation, it reached 1.95 in the presence of strain M2, indicating that the strain utilized the inorganic phosphorus in the solution for growth (Figure 1a). Compared with the treatment without strain M2 (CK treatment), inoculation with strain M2 (M2 treatment) significantly (*p* < 0.05) increased the concentration of PO_4_^3−^ (from 2.23 mg L^−1^ to 62.5 mg L^−1^) in the solution, indicating that strain M2 could dissolve calcium phosphate and release soluble PO_4_^3−^ (Figure 1b). Further testing showed that the solution pH in the CK group remained between 7.04 and 7.08, whereas the solution pH in the M2 treatment increased with cultivation time. On the 7th day of cultivation, the pH was 7.31 (Figure 1c). The HPLC results showed that strain M2 secreted ascorbic acid under Cd stress (Appendix A), which is one mechanism by which strain M2 dissolves inorganic phosphorus. The Cd concentration in the CK group remained at 4.84 mg L^−1^, while the Cd concentration in the M2 treatment significantly decreased with increasing culture time (Figure 1d). On the fifth day, the Cd concentration was 0.64 mg L^−1^, and the Cd-removal percentage reached 86.9%. Under Cd stress, strain M2 exhibited some cells with a concave shape and many white spots (containing Cd) on the cell wall according to SEM-EDS images (Figure 1e,f and Appendix A). This finding indicates that strain M2 induced Cd phosphate precipitation, thereby reducing the content of available Cd in the solution.

### 2.2. Effects of Strain M2 on the Growth of Wheat and Accumulation of Cd

Cd has a toxic effect on plant growth, and hydroponic experiments revealed that Cd inhibited wheat growth; however, this stress was relieved by the presence of strain M2 (Figure 2a,b). Compared with the CK treatment, the Cd treatment significantly (*p* < 0.05) reduced the dry weights of the wheat roots (25.9%) and leaves (41.8%), whereas the M2 strain significantly (*p* < 0.05) increased the dry weights of the wheat roots (63.1%) and leaves (36.6%) (Figure 2c). In addition, the dry weights of the wheat roots and leaves in the M2+Cd treatment group were significantly (*p* < 0.05) greater than those in the Cd treatment group (56.1%), indicating that strain M2 improved wheat resistance to Cd. The Cd contents in wheat roots and leaves in the Cd treatment were 3.51 mg kg^−1^ and 1.14 mg kg^−1^, respectively, whereas inoculation with strain M2 significantly reduced the Cd contents in wheat roots (64.6%) and leaves (64.1%) (Figure 2d), indicating that strain M2 blocks the absorption of Cd in wheat.

### 2.3. Effects of Strain M2 on Antioxidant Enzyme Activity in Wheat

Compared with the CK treatment, the Cd treatment significantly (*p* < 0.05) increased the enzyme activities of superoxide dismutase (SOD) (44.7–153%), peroxidase (POD) (28.2–87.1%), and catalase (CAT) (52.9–82.1%) in wheat roots and leaves (Figure 3). These findings indicate that in response to Cd stress, plants synthesize antioxidant enzymes to help them resist Cd toxicity. Inoculation with strain M2 did not affect the activities of SOD, POD, or CAT in wheat. However, compared with the Cd treatment, the M2+Cd treatment significantly (*p* < 0.05) reduced the activities of SOD (19.1–40.7%), POD (19.8–30.1%), and CAT (11.9–22.6%) in wheat roots and leaves, indicating that strain M2 reduced the toxic effect of Cd on wheat.

### 2.4. Effects of Strain M2 on Cd Adsorption on Wheat Roots

In the Cd treatment, strong fluorescence signals were detected both on the root surface and in the root interior. The intensity of the fluorescence signals in the root tip region was greater than that in the mature region, while the intensity of the fluorescence signals on the root surface was greater than that in the root interior, indicating that Cd was enriched mainly on the root surface and in the root tip (Figure 4a). In the M2+Cd treatment, fluorescence signals were detected only on the surface of wheat roots, and almost no fluorescence signals were detected in the root interior (Figure 4b). Moreover, the Cd content on the wheat root surface in the Cd group was 0.54 mg kg^−1^, while strain M2 significantly (*p* < 0.05) reduced (40.1%) the Cd content on the root surface (Figure 4c). These findings indicated that strain M2 can inhibit Cd entry into the interior of wheat roots. In addition, the percentages of Cd distributed in the cell wall organelles and soluble fractions of wheat roots in the Cd treatment were 33.6%, 41.5%, and 24.9%, respectively (Figure 4d). However, strain M2 significantly (*p* < 0.05) reduced the percentage of Cd in the wheat root organelles and soluble fraction and increased the percentage of Cd in the wheat root cell walls (Figure 4d), which further indicates that strain M2 had the ability to enhance the retention of Cd in the cell wall in wheat roots.

### 2.5. Effects of Strain M2 on Gene Expression in Wheat Roots

A total of 227.24 Gb of clean data (raw data for wheat root genes, BioProject ID: PRJNA1133704) were obtained, and the amount of clean data from all the samples reached at least 8.24 Gb, with a Q30 base percentage of 93.77% or greater (Appendix A). The reference genome was *Triticum_Aestivum*, the reference genome version was IWGSC, and the source was http://plants.ensembl.org/Triticum_aestivum/Info/Index (accessed on 15 June 2022). Sequencing saturation indicated that the obtained sequences were evenly distributed across the genome, indicating no bias in sequencing (Appendix A). On the basis of the expression levels, principal component analysis (PCA) was conducted on the sample clustering results. All the samples were generally divided into three clusters: an M2 cluster, an M2+Cd cluster, and a nonbacterial cluster (the CK and Cd groups) (Figure 5a). These findings indicated that strain M2 had a significant effect on gene expression in wheat roots. Sample correlation analysis confirmed that the biological repeatability of each sample was good and that the experimental design was reasonable (Figure 5b). The gene expression trends of the M2 treatment group and the M2+Cd treatment group were relatively similar. There were 46,058 genes (85.77%) shared by all treatments. The number of unique genes in the CK treatment was 726 (1.3%), the number of unique genes in the Cd treatment was 1957 (3.52%), the number of unique genes in the M2 treatment was 725 (1.3%), and the number of unique genes in the M2+Cd treatment was 523 (0.94%). In addition, 254 genes were shared by the Cd, M2, and M2+Cd groups (0.46%); 563 genes were shared by the CK, M2, and M2+Cd groups (1.01%); 1400 genes were shared by the CK, Cd, and M2+Cd groups (2.52%); and 656 genes were shared by the CK, Cd, and M2 groups (1.18%) (Figure 5c). These findings indicate that Cd had a significant impact on the expression of these genes in wheat roots. In addition, there were more upregulated DEGs than downregulated DEGs in the wheat roots, indicating that many genes related to resistance to Cd stress were expressed in wheat roots (Figure 5d). In the M2+Cd vs. Cd and M2 vs. Cd comparisons, the number of downregulated DEGs in the roots was greater than the number of upregulated DEGs (Figure 5d), indicating that strain M2 reduced the toxicity of Cd to wheat roots through its own metabolism or induced the expression of genes in wheat roots, thereby eliminating the need for wheat to overexpress certain genes.

### 2.6. Transcriptional Regulation of M2-Mediated Cd Detoxification

To quickly screen the DEGs related to Cd detoxification regulated by strain M2, the DEGs upregulated in the Cd vs. CK treatment comparison, which were related to Cd resistance, were selected. Second, the downregulated DEGs in the M2+Cd vs. Cd treatment comparison were selected because after inoculation with strain M2, the Cd stress on the wheat roots decreased, and the expression of several resistance genes decreased; both of these factors are essentially regulated by strain M2. Finally, the upregulated DEGs in the M2+Cd vs. CK treatment comparison were screened to identify DEGs actively induced by strain M2 to cope with Cd stress. A total of 47 DEGs met the above conditions (Figure 6a, Appendix A). The genes in the reference genome included genes encoding peroxidase (*POD*, 1.11.1.7), *CHS* (2.3.1.74), naringenin 3-dioxygenase (*F3H*, 1.14.11.9), naringenin 3-dioxygenase (1.14.11.9), apyrase (3.6.1.5), xylan 1,4-beta xylosidase (3.2.1.37), and pathogen-related protein 1 (*PR1*). GO enrichment analysis of the screened genes was conducted to identify the molecular functions, cellular components, and biological processes enriched by these 47 genes (Figure 6b). In the molecular function category, these genes are involved mainly in binding, catalytic activity, transcription regulator activity, and antioxidant activity. Among the cell component terms, the proteins encoded by these genes are located within the cell, in the extracellular region, in organelles, and on membranes. Among the biological processes, these genes are involved mainly in cellular processes, metabolic processes, biological regulation, and the response to stimuli. Overall, strain M2 mainly regulated the expression of genes related to binding, antioxidant activity, and the response to stimuli in wheat roots to cope with Cd stress. KEGG functional enrichment analysis of the selected genes revealed that these 47 genes were involved in flavonoid biosynthesis, phenylpropanoid biosynthesis, benzoxazinoid biosynthesis, and purine metabolism (Figure 6c). These findings indicated that strain M2 mainly enhanced the resistance of wheat roots to Cd through the activity of metabolic pathways such as flavonoid biosynthesis and phenylpropanoid biosynthesis.

### 2.7. Transcriptional Regulation of Strain M2-Mediated Cd Immobilization

To quickly screen the DEGs related to Cd immobilization regulated by strain M2, the upregulated DEGs related to Cd immobilization in the M2+Cd vs. Cd treatment comparison were selected. Second, the upregulated DEGs in the M2+Cd vs. M2 treatment comparison were selected, because inoculation with strain M2 regulated the expression of some genes, thereby inhibiting the absorption of Cd by wheat roots. Finally, the genes whose expression was upregulated in the M2+Cd group compared with the CK group were screened out, which further confirmed that the DEGs whose expression was upregulated were actively induced by strain M2 to inhibit the absorption of Cd by wheat roots. A total of 754 DEGs met the above conditions (Figure 6d, Appendix A), indicating that under Cd stress, strain M2 upregulated the expression of 754 DEGs in wheat roots to reduce their uptake of Cd. These DEGs included genes encoding the threonine protein kinase OXI1, basic endochitinase B, the threonine protein kinase OXI1 glucose S-transferase, cinnamoyl-CoA reductase, phenylalanine ammonia-lyase (*PAL*), and POD. In terms of molecular function, these DEGs were involved mainly in binding, catalytic activity, transcription regulator activity, and transporter activity (Figure 6e). Among the cell component terms, the proteins encoded by these DEGs are located within the cell, on membranes, in organelles, and in the extracellular region. Among the biological processes, the DEGs involved metabolic processes, cellular processes, biological regulation, response to stimuli, and localization. Overall, strain M2 mainly regulated the expression of genes related to binding, cellular components, and metabolic processes in wheat roots to inhibit the absorption of Cd by the roots. Moreover, KEGG functional enrichment analysis revealed that these DEGs were involved in the MAPK signaling pathway–plant, phenylalanine metabolism, phenylpropanoid biosynthesis, plant hormone signal transduction, and alpha-linolenic acid metabolism (Figure 6f). Strain M2 mainly regulated phenylalanine metabolism to inhibit the absorption of Cd in wheat roots.

### 2.8. Activity of Signaling Pathways Induced by Strain M2

All the DEGs related to Cd immobilization and Cd detoxification were analyzed for their enrichment in metabolic pathways, revealing key metabolic pathways involved in Cd immobilization and Cd detoxification. The results showed that strain M2 increased DEG enrichment in the wheat root phenylalanine metabolism and MAPK signaling pathways to inhibit the absorption of Cd (Figure 7a and Appendix A). Aromatic-L-amino–acid/L-tryptophan decarboxylase (*DDC*, 4.1.1.28), *PAL* (4.3.1.24), peroxidase (*POD*, 1.11.1.7), cinnamoyl-CoA reductase (*bglX*, 1.2.1.44), shikimate O-hydroxycinnamoyl transferase (*CCR*, 2.3.1.133), and beta-glucosidase (*HCT*, 3.2.1.21), all of which are involved in phenylalanine metabolism, were significantly expressed in the M2+Cd group. Phenylalanine is generated into coumarine, caffeic acid, lignin, and cinnamaldehyde, all of which play important roles in preventing the entry of exogenous heavy metals into crop roots [25,26]. Moreover, strain M2 upregulated the expression of OXI1, basic endochitinase B (*ChiB*), protein phase 1 L1L (*PP2C*), and threonine protein kinase OX11 (*MAPKKK17_18*) to increase the resistance of wheat roots to Cd and reduce Cd absorption (Appendix A). In addition, the *CHS* (2.3.1.74) and *F3H* (1.14.11.9) genes, which are involved in flavonoid biosynthesis, were significantly expressed and upregulated in the M2+Cd treatment group, indicating that strain M2 enhanced flavonoid biosynthesis to increase the resistance of wheat roots to Cd and promote wheat root growth (Figure 7b).

### 2.9. qRT–PCR Verification

On the basis of the DEG analysis results for the wheat roots, the expression levels of two Cd detoxification-related genes and two Cd immobilization-related genes, *POD*, *GST*, *PP2C*, and *CHS*, were determined via qRT–PCR (Figure 8). The results revealed consistency between the expression patterns of these genes and those in the transcriptomic data. For example, *POD* expression in the Cd group was 1.3-fold greater than that in the CK group. *GST* expression in the Cd group was 1.1-fold greater than that in the CK group. Similarly, *PP2C* expression in the M2+Cd group was 1.8-fold greater than that in the Cd group. *CHS* expression in the M2+Cd group was 1.6-fold greater than that in the Cd group. Therefore, the transcriptomic data were credible.

## 3. Discussion

### 3.1. Effects of Strain M2 Inoculation on the Uptake and Separation of Cd in Wheat Roots

In our study, inoculation with the phosphorus-solubilizing bacterium *Klebsiella* sp. M2 significantly reduced the absorption of Cd by wheat roots, indicating that strain M2 had the ability to inhibit Cd absorption by wheat roots. Phosphorus-solubilizing microorganisms secrete organic acids and acid phosphatases through their own metabolism, dissolving insoluble phosphates in the soil and converting them to available phosphorus for plants [27,28]. In addition, phosphorus-solubilizing microorganisms reduce heavy metal mobility and bioavailability through cell wall adsorption and extracellular precipitation, providing a technical means for the safe utilization of heavy metal-contaminated farmlands [29]. Zhang et al. [30] reported that the phosphorus-solubilizing bacterium *Burkholderia* sp. strain N3 reduced the Cd uptake of tomato (*Solanum lycopersicum* L.) and improved seedling growth. Cheng, et al. [31] reported that inoculation with *Pseudomonas taiwanensis* WRS8 significantly reduced the water-soluble Cd concentration in rhizosphere soil, thereby reducing the Cd content in wheat grains. The cell wall of phosphorus-solubilizing bacteria has been shown to immobilize a large amount of heavy metal ions, thereby reducing the content of heavy metals in the solution [32]. In the present study, strain M2 reduced the Cd content in the solution by 86.9% through the adsorption of Cd to cell walls (Figure 1). In addition, under Cd stress, strain M2 secreted ascorbic acid and induced phosphate precipitation, which further reduced the content of Cd in the solution (Figure 1 and Appendix A). This was one mechanism by which strain M2 inhibits Cd absorption in wheat.

More importantly, our observations revealed that in the M2+Cd treatment, Cd fluorescence signals were detected only on the surface of wheat roots, and almost no fluorescence signals were detected in the interior of the roots (Figure 4). The results indicate that strain M2 was adsorbed onto the wheat root surface, occupying the adsorption site of Cd on the wheat root surface and preventing Cd from entering the wheat root interior. Heavy metal ions in the soil mainly enter the root epidermal layer in three ways [33]: (1) The plant rhizosphere respiration releases CO_2_ and H_2_O, which produce H_2_CO_3_ on the cytoplasmic membrane of the root epidermis. H_2_CO_3_ dissociates from H^+^ and HCO_3_^−^, and HCO_3_^−^ can adsorb heavy metals. Heavy metal ions are adsorbed on plant root epidermal cells, which are characterized by rapid adsorption and a lack of need to provide adsorption energy. The interactive adsorption of heavy metals on plant roots provides conditions for heavy metals to enter epidermal cells [34]. (2) Nonessential heavy metals occupy the ion channels for essential heavy metals such as Fe^2+^, Zn^2+^, and Ca^2+^ [35]. (3) When organic acids are secreted from plant roots to chelate heavy metals, metal coordination complexes are formed, which enter root cells via chelation [36]. Plants can also change the morphology and distribution of their root systems under heavy metal stress, thereby reducing the content of heavy metals. Plant root tips can adsorb significant amounts on heavy metal ions in soil solutions, and different parts of the root tip respond differently to heavy metals, which affects the absorption of heavy metals by the root system [37]. Further research suggested that strain M2 increased the percentage of Cd in the root cell wall, thereby reducing the percentage of Cd in the organelles and soluble parts of wheat roots. In addition, *Ralstonia eutropha* Q2-8 increased the proportions of Cd and arsenic (As) in the wheat root cell wall and reduced the proportions of Cd and As in both soluble fractions [38]. In addition, there are many mechanisms by which phosphorus-solubilizing microorganisms dissolve insoluble phosphorus. Although strain M2 secretes organic acids to dissolve insoluble phosphorus, the content of organic acids may not be high. In addition, strain M2 may also be able to secrete some alkaline substances to increase the pH of the solution. Therefore, this is not contradictory. In subsequent studies, we plan to use metabolomics to determine the composition of the strain M2 fermentation broth and explore the specific mechanism by which strain M2 increases the pH of the solution. Overall, strain M2 reduced the uptake of Cd in wheat roots by (1) inducing phosphate-cadmium precipitation and reducing the Cd concentration in the soil solution, (2) preventing exogenous Cd from entering the wheat root interior, and (3) enhancing the adsorption capacity of wheat root cell walls for Cd.

### 3.2. Effects of Strain M2 Inoculation on Cd Detoxification in Wheat Roots

Roots, which come into direct contact with heavy metals, are also the main binding site for microorganisms on crops [16]. Functional bacteria can regulate the expression of genes related to heavy metal detoxification and immobilization in plant roots, thereby inhibiting the absorption of heavy metals by plants [24]. In the M2+Cd vs. Cd and M2 vs. Cd comparisons, the number of downregulated DEGs in the roots was greater than the number of upregulated DEGs (Figure 5d), indicating that strain M2 reduced the toxicity of Cd to wheat roots through its own metabolism or induced the expression of genes in wheat roots, thereby eliminating the need for wheat to overexpress certain genes. Strain M2 mainly alleviated the toxicity of Cd to wheat roots by regulating the antioxidant enzyme system and flavonoid biosynthesis.

To combat the toxicity of heavy metals, plants have evolved several effective defense mechanisms, including SOD, CAT, APX, and POD activities [39,40]. Strain M2 increased the expression of *POD* in wheat roots. Cd inhibits the antioxidant defense system of wheat plants and triggers the formation of reactive oxygen species (ROS), thereby inducing oxidative stress and ultimately leading to cell apoptosis [41]. POD enzymes, including catalase and peroxide reductase, catalyze the decomposition of ROS and protect cells from oxidative damage [42]. Peroxidase reductases are a protein family whose members play important roles in ROS clearance and redox signal transduction and have been proven to protect cells from oxidative damage [43]. Our physiological and biochemical data also showed that strain M2 increased POD activity in wheat roots, which was consistent with the transcriptomic results.

Flavonoids, as effective endogenous regulators of plant auxin, are widely involved in plant growth and development [44]. In addition to regulating plant growth and development, flavonoids play a role in the response to environmental stress [45]. KEGG enrichment analysis revealed that inoculation with strain M2 significantly increased flavonoid biosynthesis (Figure 6), although the expression levels of genes such as *CHS* and *F3H* were also significantly greater than those in the CK group (Figure 7). Li et al. [46] reported that flavonoid and metal transport-related protein-coding genes promoted potato resistance to Cd. *CHS* and *F3H*, which are rate-limiting enzymes for the synthesis of flavonoids, and anthocyanins were also significantly upregulated in the M2+Cd group (Figure 7). Flavonoids exert their antioxidant effects through two main mechanisms: (1) As defensive antioxidants, they have strong free radical scavenging and antioxidant activity and hinder free radical chain reactions in the form of free radical acceptors, and (2) they chelate with heavy metal ions to slow oxidative damage to roots. When plants are subjected to Cd stress, their cells induce the regulation of metal ion homeostasis and the aggregation of nonenzymatic antioxidant flavonoids, ensuring that plant cells are protected from oxidative damage caused by ROS [47].

### 3.3. Effects of Strain M2 Inoculation on Cd Immobilization in Wheat Roots

Compared with those in the Cd group, the phenylalanine metabolic pathway and MAPK signaling pathway-related genes were significantly enriched in the M2+Cd group, indicating that these metabolic pathways are involved in the immobilization of Cd by wheat roots (Figure 6 and Figure 7). The metabolites of the phenylpropanoid pathway scavenge ROS and slow membrane peroxidation and thus play an important antioxidant role in plant resistance to heavy metal stress [48]. Lignin is a complex phenolic compound formed by further transport and polymerization of lignin monomers synthesized through the phenylpropane metabolic pathway. The main function of lignin is to provide a structural and defensive barrier for cell walls, which is closely related to plant stress resistance [49]. In our study, the expression levels of genes involved in lignin biosynthesis, such as *PAL*, *CCR*, and *DDC*, were significantly greater in the M2+Cd group than in the CK group (Figure 7). The cell wall is a key site for the storage of heavy metal ions in plants and the first barrier for plants to absorb and transport heavy metal ions [25,50]. The mechanism of Cd toxicity in wheat begins with exposure to the root cell wall. Plants utilize ROS-dependent lignification to bind heavy metal ions to functional groups such as lignin carboxyl, phenolic, and aldehyde groups, thereby immobilizing heavy metal ions in the cell wall, reducing their ability to migrate into the plant body, inhibiting heavy metal ions from entering the cytoplasm, and enhancing plant tolerance to heavy metal stress [51]. Ge and Li [26] found that the heavy metal ion adsorbent obtained from lignin had good stability, biocompatibility, and adsorption ability; therefore, lignin and its derivatives could be converted into heavy metal ion adsorbents for environmental treatment. Wang et al. [52] reported that the application of lignin increased the growth rate of wheat (*Triticum aestivum* L.) and reduced the accumulation of heavy metals in wheat. The application of lignin increased the content of soil organic matter, effectively adsorbed free cations in soil solutions, and reduced the mobility and bioavailability of heavy metals in the soil.

Phytohormones also play important roles in mediating the absorption of heavy metals by plants [53]. In this study, strain M2 promoted the upregulation of genes related to abscisic acid (ABA) and ethylene synthesis under Cd stress, indicating that ABA and ethylene are involved in the immobilization of Cd in wheat roots. Owing to its important role in the response to various environmental stresses, ABA is known as a “stress hormone”. In plants, ABA plays an important role in seed germination, plant growth, and response to environmental stresses [40,54]. Under Cd stress, the ABA content in Cd-tolerant rice varieties was significantly greater than that in conventional rice varieties [55]. The application of exogenous ABA inhibited the accumulation of Cd in *Arabidopsis* [56]. Ethylene is an important plant growth hormone that participates in several physiological processes, such as senescence, leaf abscission, and fruit ripening. [57]. Numerous studies have shown that ethylene can alleviate the harm caused by heavy metals to crop plants and inhibit their absorption of Cd [58,59]. Guan et al. [60] revealed the protective effect of ethylene on Cd tolerance, and the upregulation of the LchERF gene and accumulation of glutathione (GSH) increased endogenous ethylene production in plants, thereby increasing their resistance to Cd. In general, strain M2 affected the expression levels of ABA- and ethylene-related genes in wheat roots, thereby increasing resistance to Cd.

## 4. Materials and Methods

### 4.1. Phosphorus-Solubilizing Bacteria

The phosphorus-solubilizing bacterium M2 (CCTCC M 2024654), which was isolated from heavy metal-contaminated radish rhizosphere soil, can secrete organic acids to dissolve insoluble phosphorus [13]. In a solution with 3 mg L^−1^ Cd, the Cd-removal percentage reached 70.3%. The seed solution (0.3 mL) was inoculated into 10 mL Luria–Bertani liquid medium (5.0 g L^−1^ beef extract, 10.0 g L^−1^ peptone, and 5.0 g L^−1^ NaCl; pH 7.0) and incubated at 30 °C for 24 h. The culture was subsequently centrifuged, after which the bacterial cells were collected. Finally, the bacterial cells were resuspended in deionized water, and the OD_600_ of the solution was adjusted to 1.0. All the experiments were conducted in the Laboratory of Heavy Metal Pollution Control and Element Turnover in the Experimental Building of Nanyang Normal University.

### 4.2. Cd Immobilization Effect of Strain M2

A 7-day dynamic shake-flask experiment was performed to investigate the effect and mechanism of Cd adsorption by strain M2. Inorganic phosphorus culture medium (3.0 g Ca_3_(PO_4_)_2_, 5.0 g yeast extract, 10 g sucrose, 0.3 g NaCl, 0.03 g MnSO_4_•H_2_O, 0.3 g KCl, 0.03 g FeSO_4_•7H_2_O, 0.5 g (NH_4_)_2_SO_4_, 0.3 g MgSO_4_•7H_2_O, and 1 L deionized water; pH 7.0) containing 5 mg L^−1^ Cd (Cd(NO_3_)_2_) was prepared, and 50 mL aliquots of the culture medium were added to triangular flasks [13]. The suspension of strain M2 was inoculated into a triangular flask such that the concentration was 1% (M2), along with a non-inoculated control (CK). A portion of the fermentation broth was removed (7 mL) on days 0, 1, 3, 5, and 7. The OD_600_ of the culture medium was determined by spectrophotometry, the pH of the culture medium was determined by a pH meter, the concentration of Cd in the culture medium was determined by inductively coupled plasma–optical emission spectrometry (ICP–OES) [61], and the content of PO_4_^3−^ was determined by spectrophotometry [62]. In addition, a 7-day solution adsorption test involving the M2+Cd (5 mg L^−1^ Cd) and M2 treatment groups was performed. After cultivation, the culture medium was centrifuged at 5000 rpm to collect the supernatant and bacterial precipitates. Liquid chromatography was used to determine the types of organic acids in the supernatant [63]. After freeze drying, the bacterial precipitates were observed by scanning electron microscopy coupled with energy dispersive X-ray (SEM-EDX) to determine their morphology [64].

### 4.3. Wheat Cultivation Experiment

Several seeds of wheat (*Triticum aestivum* L., JiMai-22) were selected, soaked in 75% ethanol for 2 min, and then washed in sterile deionized water 3 times. Thirty wheat seeds were sown on the upper layer of a pot, and 1.7 L of Hoagland culture medium (945 mg L^−1^ (Ca(NO_3_)_2_, 607 mg L^−1^ KNO_3_, 115 mg L^−1^ NH_4_•H_2_PO_4_, 493 mg L^−1^ MgSO_4_•7H_2_O, 2.13 mg L^−1^ MnSO_4_, 0.08 mg L^−1^ CuSO_4_, 0.22 mg L^−1^ ZnSO_4_, 2.86 mg L^−1^ H_3_BO_3_, 0.02 mg L^−1^ H_2_MoO_4_, and 40 mg L^−1^ chelated iron; pH 6.8) was added to the lower layer of the pot. There were four treatments in the experiment: a CK treatment, 1 mg L^−1^ Cd treatment (Cd), strain M2 treatment (M2), and M2+Cd treatment (M2+Cd). Each treatment had three replicates. When the wheat seedlings had grown to a height of 8 cm, inoculation with strain M2 was carried out by soaking the roots in the inoculant, and the wheat seedlings were subsequently moved to a pot containing a suspension of strain M2 and soaked for 2 h. The experiment was conducted in a greenhouse (20 °C), and the nutrient solution was changed every other week; the Cd solution and bacterial suspension were also replenished. After the Cd solution was added, the wheat plants were harvested after cultivation for 2 weeks.

### 4.4. Determination of Wheat Root and Leaf Dry Weights and Cd Content

The wheat plants were removed from the pots and divided into roots and leaves. These materials were placed in a Na_2_-EDTA (0.1 mol L^−1^) solution and rinsed for 10 min. Then, the wheat roots and leaves were rinsed 3 times with sterile deionized water, placed in an oven (60 °C), and dried until a constant weight was reached. The dry weights of the wheat roots and leaves were measured via a balance, and the Cd content was determined via ICP-OES. Moreover, to investigate the changes in enzyme activity in fresh wheat tissues under Cd stress, a reagent kit was used to measure POD, SOD, and CAT activities [65].

### 4.5. Determination of the Cd Content on the Surface of Wheat Roots

A Cd fluorescence probe was dissolved in 50 μL of dimethyl sulfoxide (DMSO) and diluted with a 0.85% NaCl solution at a ratio of 1:10. The wheat root tips (1 cm in length) were subsequently immersed in a dye solution for 1.5 h, followed by washing with a detergent containing 0.85% NaCl and 1 mmol L^−1^ Na_2_-EDTA for 24 h. The roots were placed on a glass slide and observed using a Zeiss laser confocal microscope. The wheat roots (0.5 g) were placed into a 15 mL centrifuge tube. Five milliliters of extraction solution (0.25 mol L^−1^ NaOH and 0.05 mol L^−1^ Na_2_-EDTA) was added to the centrifuge tube to extract the surface Cd. After allowing sufficient time for extraction, the extraction solution was transferred to a 50 mL polytetrafluoroethylene tube, and 6 mL an acid mixture (HNO_3_:HCl = 4:1) was added. The polytetrafluoroethylene tube was heated at 200 °C to dry the acid nearly completely. The Cd content in the extraction mixture was determined after the samples were filtered through a membrane.

### 4.6. Cd Subcellular Distribution in Wheat Roots

Fresh wheat roots (0.5 g) were homogenized in a precooled solution (250 mmol L^−1^ sucrose, 1.0 mmol L^−1^ dithiothreitol (DTT), and 50 mmol L^−1^ Tris-HCl (pH 7.5)). The cell wall, organelle and soluble fractions of the wheat roots were separated by differential centrifugation. All homogenization and separation processes were performed at 4 °C. Fifteen milliliters of homogenate solution was then transferred to a 50 mL centrifuge tube and centrifuged at 300 r min^−1^ for 30 s. The sediment in the lower layer was considered the cell wall. The upper layer was subsequently centrifuged at 20,000 r min^−1^ for 45 min, after which the lower layer was precipitated and considered the organelle component, with the supernatant or soluble fraction, which included the cytoplasm and vacuoles. The Cd content in each component was determined via ICP–OES.

### 4.7. Determination of Differentially Expressed Genes in Wheat Roots

Fresh wheat roots were washed with 0.1% diethyl phosphorocyanidate (DEPC) water to remove surface dirt, dried on filter paper, and then cut into small (0.5 cm long) pieces. The wheat roots were placed in a precooled RNase-free cryopreservation tube and frozen in liquid nitrogen for 0.5 h. An MJzol Total RNA Extraction Kit was used to extract the wheat root RNA. The detailed RNA extraction and sequencing steps are described in the Appendix A. The screening criteria for DEGs included a false-discovery rate (FDR) < 0.05 and a |log2(fold change (FC)| ≥ 1. GOATOOLS (https://github.com/tanghaibao/GOatools, accessed on 28 June 2022) was used to perform a gene ontology (GO) enrichment analysis, and KOBAS (http://kobas.cbi.pku.edu.cn/home.do, accessed on 12 July 2022) was used to conduct Kyoto Encyclopedia of Genes and Genomes (KEGG) pathway enrichment analysis of the DEGs [66]. All the analyses were conducted on the Meiji Cloud platform (https://cloud.majorbio.com/, accessed on 12 July 2022) [67].

### 4.8. Quantitative Real-Time PCR (qRT–PCR) Verification

To verify the accuracy of the transcriptome data, the Cd detoxification-related genes encoding POD (primers: 5′-CTTCTACCAGGTCCCTTCCG-′3 and 5′-GTTGGTGAACTGGCTCGTGT-′3) and GST (primers: 5′-TGGACGAACTTTTCTGCTTTTT-′3 and 5′-CAGCCGCTCCTC ATAGACCT-′3) and the Cd immobilization-related genes encoding PP2C (primers: 5′-ATCCGACCACA GGTATGCG-′3, 5′-TCGGSTATCACGAATGGGCTT-′3) and chalcone synthase (CHS) (primers: 5′-GCCTCACCTCCATTCATCTCCTC-′3, 5′-CGTTTACGCATCTCATCCAG-′3) were selected for qRT–PCR analysis. qRT–PCR was performed on a LightCycler 480 II (Roche, Basel, Switzerland) real-time system. The relative expression levels were determined according to the methods of Fisher et al. [68].

### 4.9. Statistical Analysis

All the statistical analyses were performed via Excel 2010 and SPSS 20.0. The mathematically processed results are presented as M ± SE, where M is the arithmetic mean, and SE is the standard error (*n* = 3). One-way analysis of variance (ANOVA) and Duncan’s honestly significant difference (HSD) multiple comparison test were used to analyze the test data. Transcriptome data analysis was subsequently conducted on the Meiji Cloud platform (https://cloud.majorbio.com/, accessed on 12 July 2022).

## 5. Conclusions

Phosphorus-solubilizing bacterium M2 reduced the accumulation of Cd in wheat roots and leaves. The possible mechanisms were as follows: (1) Strain M2 induced Cd phosphate precipitation and reduced the concentration of Cd in the solution; (2) strain M2 prevented exogenous Cd from entering the wheat root interior; (3) strain M2 increased the adsorption capacity of wheat root cell walls for Cd; (4) strain M2 regulated the activity of the phenylalanine metabolism and MAPK signaling pathways in wheat roots and increased the expression levels of genes (*PAL*, *CCR*, and *DDC*) related to Cd immobilization; and (5) strain M2 regulated the flavonoid biosynthesis metabolic pathway in wheat roots and increased the expression of genes related to Cd detoxification (such as *POD*, *CHS*, and *F3H*), which play a significant role in alleviating Cd toxicity. Further in-depth research can be conducted on genes such as *CHS* and *F3H* to increase the resistance to and immobilization of Cd in wheat roots to develop wheat varieties with low Cd accumulation.

## Figures and Tables

**Figure 1 plants-13-01989-f001:**
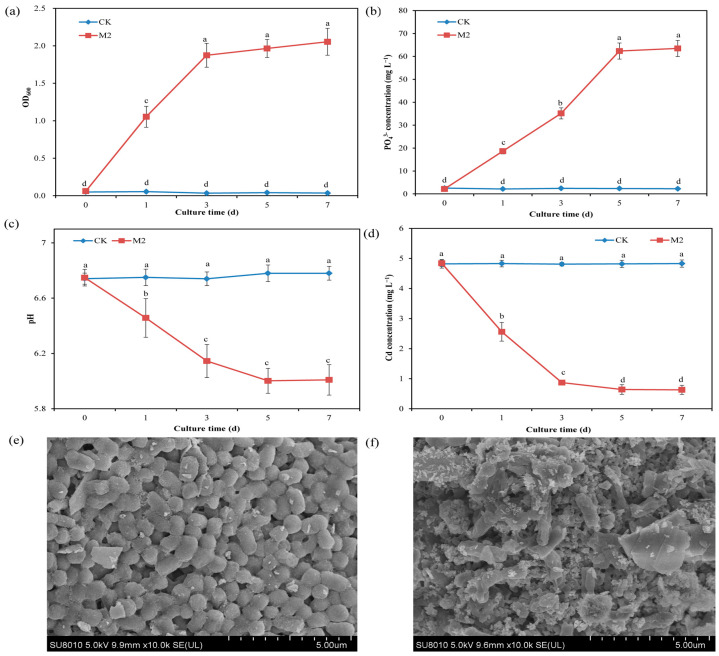
Effect of M2 on the immobilization of Cd in a solution containing 5 mg L^−1^ Cd. (**a**) OD_600_; (**b**) PO_4_^3−^ concentration; (**c**) pH; (**d**) Cd^−^ concentration; (**e**) SEM image of strain M2 in the absence of Cd; (**f**) SEM image of strain M2 in the presence of 5 mg L^−1^ Cd. The values are presented as the means and standard deviations (*n* = 3), and indexes with different lowercase letters are significantly different at *p* < 0.05.

**Figure 2 plants-13-01989-f002:**
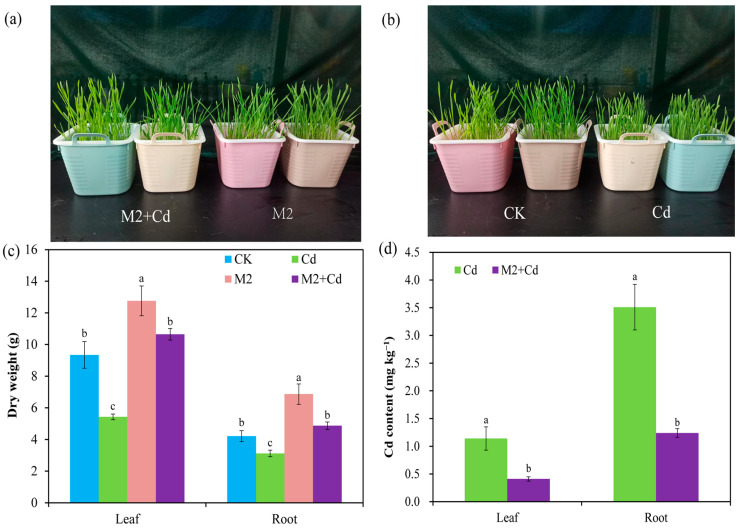
Biomass of wheat and changes in the CD content in the presence of strain M2 and Cd. (**a**) Comparison of wheat growth under the M2+Cd and M2 treatments; (**b**) comparison of wheat growth under the CK and Cd treatments; (**c**) dry weight under the different treatments; (**d**) Cd content under the different treatments. The error bars represent ± standard errors (*n* = 3). The bars indicated by different letters within each wheat tissue (root and leaf) indicate significant differences (*p* < 0.05) according to one-way ANOVA.

**Figure 3 plants-13-01989-f003:**
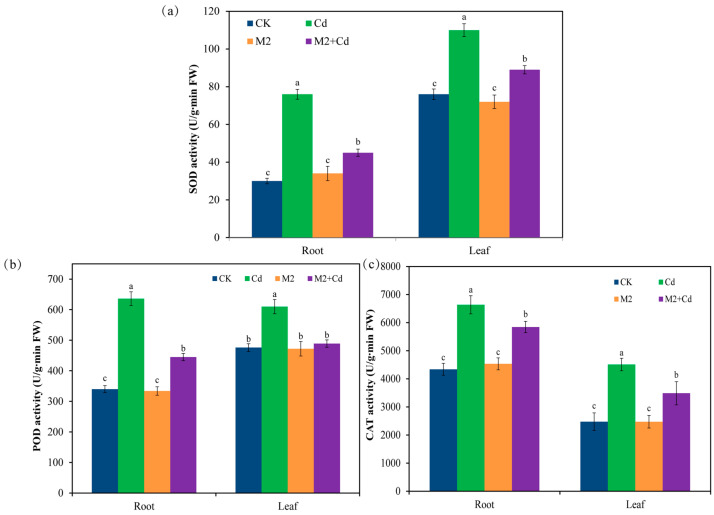
Effects of different treatments on antioxidant enzyme activities in wheat roots and leaves. (**a**) SOD activity; (**b**) POD activity; (**c**) CAT activity. The error bars represent ± standard errors (*n* = 3). The bars indicated by different letters within each wheat tissue (root and leaf) indicate significant differences (*p* < 0.05) according to one-way ANOVA.

**Figure 4 plants-13-01989-f004:**
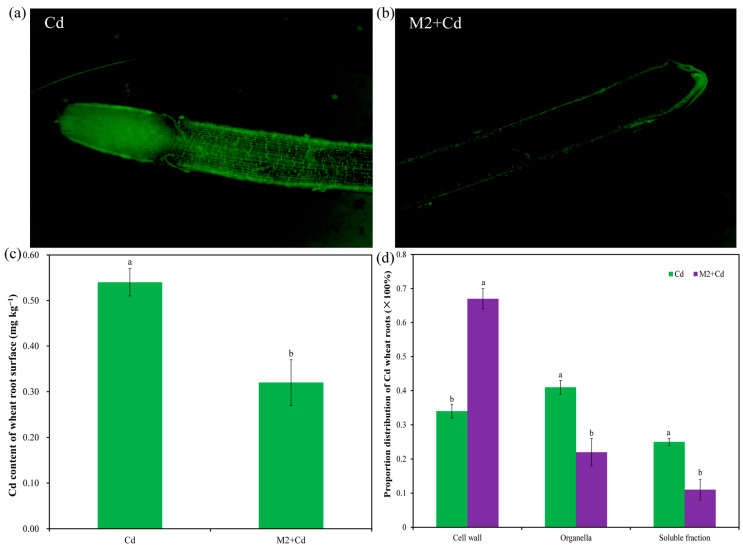
Effects of strain M2 on Cd adsorption on the wheat root surface. (**a**) Cd fluorescence probe staining diagram in the presence of 1 mg L^−1^ Cd; (**b**) Cd fluorescence probe staining diagram in the presence of strain M2 and 1 mg L^−1^ Cd; (**c**) Cd content in wheat root surface; (**d**) distribution proportion of Cd in the wheat roots. The error bars represent ± standard errors (*n* = 3). The bars indicated by different letters within each wheat tissue indicate significant differences (*p* < 0.05) according to one-way ANOVA.

**Figure 5 plants-13-01989-f005:**
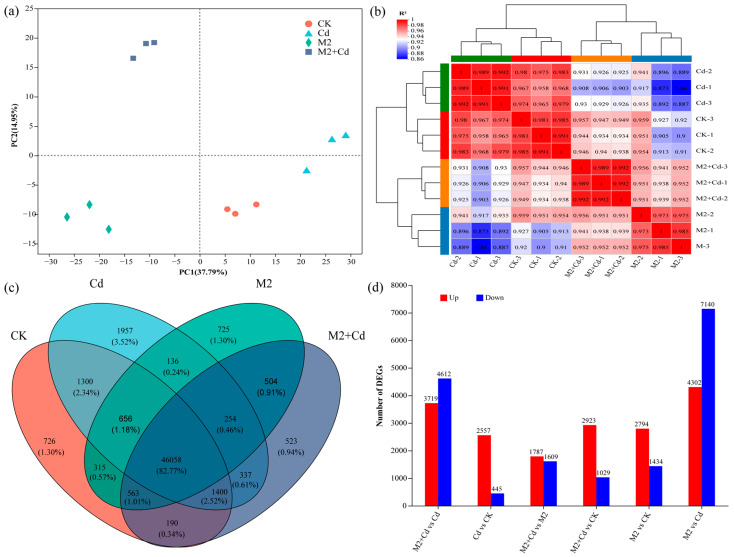
Effects of strain M2 on gene expression in wheat roots. (**a**) PCA of different samples; (**b**) correlation analysis of different samples; (**c**) Venn diagrams of DEGs; (**d**) number of genes differentially expressed between different treatments.

**Figure 6 plants-13-01989-f006:**
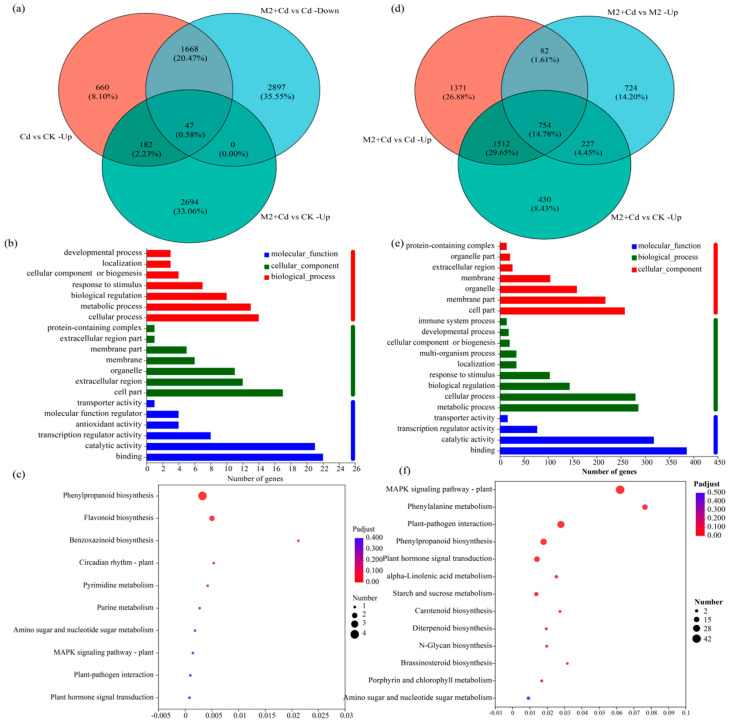
Effects of strain M2 on Cd detoxification and Cd immobilization related gene expression in wheat roots. (**a**) Venn diagrams of DEGs involved in Cd detoxification; (**b**) GO enrichment analysis of Cd detoxification genes; (**c**) KEGG enrichment analysis of Cd detoxification genes; (**d**) Venn diagrams of DEGs involved in Cd immobilization; (**e**) GO enrichment analysis of Cd immobilization genes; (**f**) KEGG enrichment analysis of Cd immobilization genes.

**Figure 7 plants-13-01989-f007:**
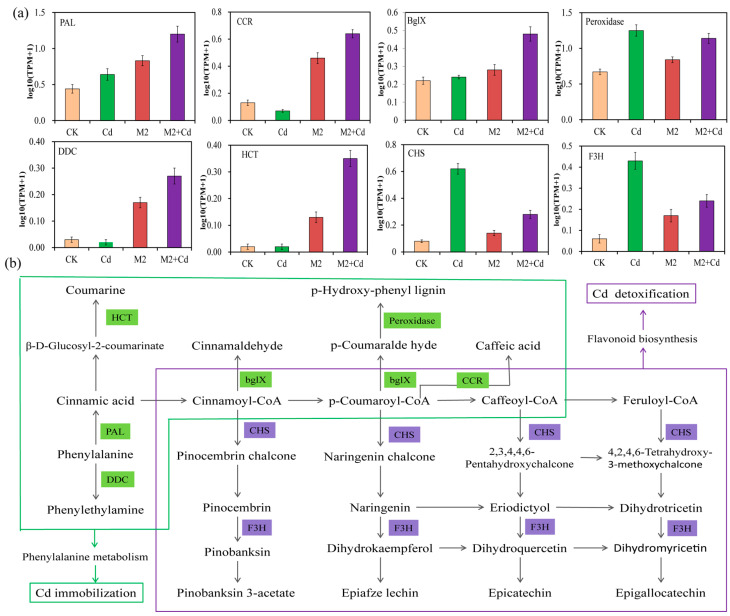
Key metabolic pathways in wheat roots for Cd immobilization and detoxification mediated by strain M2. (**a**) Transcripts per kilobase million (TPM) values of genes in the samples; (**b**) key metabolic pathways related to Cd immobilization and detoxification. DDC: aromatic-L-amino-acid/L-tryptophan decarboxylase; PAL: phenylalanine ammonia-lyase; HCT: beta-glucosidase; bglX: cinnamoyl-CoA reductase; CCR: shikimate O-hydroxycinnamoyl transferase; CHS: chalcone synthesis; F3H: naringenin 3-dioxygenase.

**Figure 8 plants-13-01989-f008:**
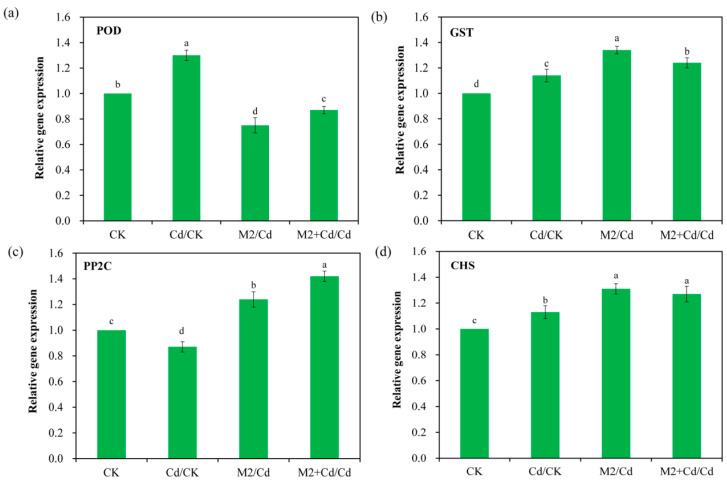
Gene expression levels were determined by qRT–PCR. CK: wheat roots not treated with Cd and not inoculated with strain M2; Cd: wheat roots treated with 1 mg L^−1^ Cd; M2: wheat roots inoculated with strain M2; and M2+Cd: wheat roots treated with 1 mg L^−1^ Cd and inoculated with strain M2. The data are presented as the means ± SD (*n* = 3). The bars indicated by different letters are significantly different (*p* < 0.05) according to one-way ANOVA.

## Data Availability

The raw data supporting the conclusions of this article will be made available by the authors, without undue reservation. The original sequencing data generated in the study have been deposited into the National Center for Biotechnology Information (NCBI) Sequence Read Archive (SRA) database with accession number PRJNA1133704.

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
