# Peer review of "Phosphorus-Solubilizing Bacteria Enhance Cadmium Immobilization and Gene Expression in Wheat Roots to Reduce Cadmium Uptake"

_plants, 2024, doi:10.3390/plants13141989_

Round 1

Reviewer 1 Report

Comments and Suggestions for Authors

Specific comments:

1) The title of the manuscript should be modify.

2) The English language of the MS needs minor checks. There are some  occasional ambiguity in the flow of the sentence, mostly in abstract section.

3) Rewrite the abstract in a systamtic flow in line with the research. The strain name should be written in full.

4) Authors should clearly define so many abbreviations in the text.

5) Only vertical and horizontal lines should be presented in Figure 1a – 1d, Figure 2c and 2d, Figure 3a to 3c, Figure 4c and 4d, and Figure 7a. Check other figures as well

Minor comments.

L4: write Cd in full. Also in L16, write in full and bracket the abbreviation at the first mentioning.

L527: Transcriptomice???

Comments on the Quality of English Language

Minor checks

Author Response

Comments 1: The title of the manuscript should be modify.

Response 1: Thank you for pointing this out. We agree with this comment. Therefore, we have revised the title. The new title is “Phosphorus-solubilizing bacteria enhance cadmium immobilization and gene expression in wheat roots to reduce cadmium uptake”

Comments 2: The English language of the MS needs minor checks. There are some occasional ambiguity in the flow of the sentence, mostly in abstract section.

Response 2: Thank you for pointing this out. We agree with this comment. The language of this manuscript has been revised and improved by native English speakers (https://www.aje.cn/).

Comments 3: Rewrite the abstract in a systamtic flow in line with the research. The strain name should be written in full.

Response 3: Thank you for pointing this out. We agree with this comment. Therefore, we have revised the abstract.

The application of phosphorus-solubilizing bacteria is an effective method for increasing the available phosphorus content and inhibiting wheat uptake of heavy metals. However, further research is needed on the mechanism by which phosphorus-solubilizing bacteria inhibit cadmium (Cd) uptake in wheat roots and its impact on the expression of root-related genes. Here, the effects of strain Klebsiella aerogenes M2 on Cd absorption in wheat and the expression of root-related Cd detoxification and immobilization genes were determined. Compared with the control, strain M2 reduced (64.1%-64.6%) Cd uptake by wheat roots. Cd fluorescence staining revealed that strain M2 blocked the entry of exogenous Cd into the root interior and enhanced the immobilization of Cd by cell walls. Forty-seven genes related to Cd detoxification, including genes encoding peroxidase, chalcone synthase, and naringenin 3-dioxygenase, were upregulated in the Cd+M2 treatment. Strain M2 enhanced the Cd resistance and detoxification activity of wheat roots through the regulation of flavonoid biosynthesis and antioxidant enzyme activity. Moreover, strain M2 regulated the expression of genes related to phenylalanine metabolism and MAPK signaling pathway to enhance Cd immobilization in roots. These results provide a theoretical basis for the use of phosphorus-solubilizing bacteria to remediate Cd-contaminated fields and reduce Cd uptake in wheat.

Comments 4: Authors should clearly define so many abbreviations in the text.

Response 4: Thank you for pointing this out. We agree with this comment. Therefore, we have provided explanations and explanations for all abbreviations in the paper.

Comments 5: Only vertical and horizontal lines should be presented in Figure 1a – 1d, Figure 2c and 2d, Figure 3a to 3c, Figure 4c and 4d, and Figure 7a. Check other figures as well

Minor comments.

Response 5: Thank you for pointing this out. We have revised all figures as as suggested.

Comments 6: L4: write Cd in full. Also in L16, write in full and bracket the abbreviation at the first mentioning.

Response 6: Thank you for pointing this out. We agree with this comment. Therefore, we have revised the abbreviations at the first mention.

Comments 7: L527: Transcriptomice???

Response 7: Thank you for pointing this out. We have revised the text to “Determination of differentially expressed genes in wheat roots.

Reviewer 2 Report

Comments and Suggestions for Authors

Manuscript plants-2998862 presents interesting data on common wheat under conditions of high cadmium concentrations. However, the presentation of the data and the discussion of the results raise many criticisms. In this regard, I believe that this manuscript can be accepted for publication after corrections.

Use “cadmium” rather than “Cd” in the manuscript title, keywords, and first mention in the abstract.

Lines 36–39 should contain references. Reference [4] describes results for durum wheat, while the manuscript describes results for soft wheat. Unexpectedly, “Wheat (Triticum aestivum L.) is the main food crop in China...” - according to FAO, wheat production in China is 1.5 times inferior to rice and 2 times inferior to maize.

Line 58 and down: Have the authors deposited the M2 strain in any collection of microorganisms? If so, this should be indicated in section 4.1. If not, you should do so. Work with bacterial strains that are not stored in collections does not have any scientific significance. I also have a note regarding the name of the strain: The authors call the strain Klebsiella sp. M2, referring to the work [14]. However, the article by Qin et al., 2023 provides an extremely weak and inconclusive taxonomic description of the bacterial strain. The specified accessory number OP090254 is inactive in GenBank, so I (as a reviewer) have no reason to consider strain M2 as Klebsiella sp. M2.

Lines 91-92: “The solution adsorption test showed that strain M2 could grow well in inorganic phosphorus medium.” - It should be stated that the source of phosphorus was water-insoluble calcium phosphate.

Lines 92-93: “On the second day of cultivation, the OD600 was 1.11, and on the 92 third day of cultivation, …” - these data do not correspond to Figure 1a.

Line 94: When using the abbreviation “CK” for the first time, authors should clarify the meaning.

Lines 97-99: Here I see some contradiction - the authors previously wrote that in order to dissolve calcium phosphate, bacteria must secrete organic acids, then it will be indicated that the M2 strain produces ascorbic acid into the medium, but the pH of the medium increases. Why? There is no explanation for this in the manuscript. How important is a 0.2 unit increase in pH?

Line 106: “white spots (containing Cd)” - Based on what data do the authors believe that the white spots contain cadmium?

Line 108: How was the content of “free Cd” determined? Do the authors consider cadmium cations bound to organic acids to also be “free Cd”?

Lines 128-129: “other 128 researchers have reached similar conclusions [29,30].” - this information should be moved to the Discussion section.

Line 145: Why do the authors interpret a decrease in enzyme activities (SOD, POD, and CAT) as “improved wheat resistance to Cd”.

Figure 3: Why is the one-way ANOVA done without dividing into “Root” and “Leaf” groups, as in Figures 2c and 2d?

Figures 4c and 4d: Provide the results of the statistical analysis of the results.

Line 176: “A total of 227.24 Gb of clean data were obtained” - Explain what data you mean.

Lines 192-194: “In addition, six comparison groups were established: M2+Cd vs. Cd, Cd vs. CK, M2+Cd vs. M2, M2 vs. Cd, M2+Cd vs. CK, and M2 vs. CK." - This proposal can be deleted since all possible pairs are indicated, and they are discussed further.

Figure 5c: The authors should explain in more detail in the caption to the figure and in the text of the manuscript what the numbers indicated in the figure mean. In particular, explain what 726 (1.30%) means for "CK". This is 726 DEGs relative to which option? If these are 726 genes, the expression of which is represented only in the control variant, then how do they differ from the 254 (0.46%) DEGs for the three variants without control? Overall, the discussion of Figure 5c is very brief, leaving the interpretation of most of the values ​​in this figure unclear. Perhaps it would be clearer to show two figures instead of Figure 5c: a diagram for up-regulated genes and a diagram for down-regulated genes.

Figure 7: The authors should improve the quality (contrast) of the drawing so that the inscriptions can be read.

Section 2.9 contains a figure that does not have a link in the text or a caption.

Line 321: “tomato (Solanum tuberosum L).” Are the authors talking about tomatoes or Solanum tuberosum L.?

Line 328: Do the authors use the term “nutrients” to refer to cadmium?

Line 340: Explain what is meant by “dissociated H+ can adsorb heavy metals.” How do hydrogen cations interact with metal cations???

Lines 358–359: Based on what data are conclusions 2 and 3 drawn?

In general, the manuscript fragment from lines 332 to 359 lacks references to the work of other authors.

Lines 366-369: The proposal is not consistent with the results presented.

Line 376: What evidence shows that strain M2 increased the expression of glutathione reductase (GR)?

Line 444: What is GSG?

Lines 452-453: To what volume of LB medium was 0.3 mL of the seed solution added?

Line 464: What cadmium compound was used in the experiments?

Line 479: Write the scientific name of common wheat (Triticum aestivum L.).

Lines 481-484: Explain what “the upper layer of a pot” and “the lower layer of a pot” mean.

Author Response

Comments 1: Use “cadmium” rather than “Cd” in the manuscript title, keywords, and first mention in the abstract.

Response 1: Thank you for pointing this out. We agree with this comment. Therefore, we have made this correction as suggested.

Comments 2: Lines 36–39 should contain references. Reference [4] describes results for durum wheat, while the manuscript describes results for soft wheat. Unexpectedly, “Wheat (Triticum aestivum L.) is the main food crop in China...” - according to FAO, wheat production in China is 1.5 times inferior to rice and 2 times inferior to maize.

Response 2: Thank you for pointing this out. We agree with this comment. Therefore, we have added some references and deleted reference [4].

Wheat (Triticum aestivum L.) is the main food crop in China and is widely cultivated [3]. Wheat plants mainly absorb Cd through their roots, where the element accumulates in the plant body, affecting cell structure, photosynthesis and enzyme activity and strongly inhibiting crop growth [4]. Moreover, Cd levels in wheat grains exceeding the national standard can have harmful effects on human health [5].

Comments 3: Line 58 and down: Have the authors deposited the M2 strain in any collection of microorganisms? If so, this should be indicated in section 4.1. If not, you should do so. Work with bacterial strains that are not stored in collections does not have any scientific significance. I also have a note regarding the name of the strain: The authors call the strain Klebsiella sp. M2, referring to the work [14]. However, the article by Qin et al., 2023 provides an extremely weak and inconclusive taxonomic description of the bacterial strain. The specified accessory number OP090254 is inactive in GenBank, so I (as a reviewer) have no reason to consider strain M2 as Klebsiella sp. M2.

Response 3: Thank you for pointing this out. We agree with this comment. Therefore, we have deposited strain M2 in the China Center for Type Culture Collection (CCTCC). Due to the time required for the preservation of bacterial strains, when we receive the proof of preservation, we added this information to section 4.1.

The identification of strain M2 was a preliminary work (Qin et al., 2023). When isolating and identifying microorganisms, the usual approach is to extract the total DNA of the strain, amplify 16S RNA, compare the at https://www.ncbi.nlm.nih.gov/ or https://www.ezbiocloud.net/ , find the strain with the highest similarity (similarity greater than 99%), and identify it as this bacterial genus. This identification method has broad scientific consensus at the genus level. Therefore, we identified strain M2 as Klebsiella sp. In addition, when we uploaded the 16S RNA information of strain M2 to NCBI, it may not have been set properly, resulting in the information not being made public. We will contact NCBI as soon as possible to active the information of strain M2.

Comments 4: Lines 91-92: “The solution adsorption test showed that strain M2 could grow well in inorganic phosphorus medium.” - It should be stated that the source of phosphorus was water-insoluble calcium phosphate.

Response 4: Agree. We have done as suggested. The solution adsorption test showed that strain M2 could grow well in inorganic phosphorus-containing medium, in which the source of phosphorus was water-insoluble calcium phosphate.

Comments 5: Lines 92-93: “On the second day of cultivation, the OD600 was 1.11, and on the 92 third day of cultivation, …” - these data do not correspond to Figure 1a.

Response 5: Thank you for pointing this out. We agree with this comment. Therefore, we have revised the text as suggested.

On the first day of cultivation, the OD600 was 1.11, and on the third day of cultivation, it reached 1.95 in the presence of strain M2, indicating that the strain utilized the inorganic phosphorus in the solution for growth (Fig. 1a).

Comments 6: Line 94: When using the abbreviation “CK” for the first time, authors should clarify the meaning.

Response 6: Agree. We have done as suggested. Compared to the treatment without inoculation with strain M2 (CK treatment), inoculation with strain M2 (M2 treatment) significantly (P<0.05) increased the concentration of PO43- (from 2.23 mg L-1 to 62.5 mg L-1) in the solution.

Comments 7: Lines 97-99: Here I see some contradiction - the authors previously wrote that in order to dissolve calcium phosphate, bacteria must secrete organic acids, then it will be indicated that the M2 strain produces ascorbic acid into the medium, but the pH of the medium increases. Why? There is no explanation for this in the manuscript. How important is a 0.2 unit increase in pH?

Response 7: The solubilization of insoluble phosphorus by phosphate-solubilizing microorganisms mainly includes the following mechanisms: (1) in acid hydrolysis, organic acids, such as lactic acid, hydroxyacetic acid, fumaric acid, and succinic acid, convert insoluble phosphorus to soluble phosphorus. (2) Enzymolysis occurs when microorganisms produce phosphorus-solubilizing enzymes, such as phosphatase, phytase, nuclease, anddehydrogenase, that can be secreted during their growth into the extracellular space. (3) Phosphate-solubilizing bacteria release H2S as a metabolic byproduct of microbial organic matter decomposition and sulfate reduction and H2S then reacts with ferric-phosphate and produces ferrous sulfate and soluble phosphate. (4) Phosphorus-solubilizing bacteria decompose plant residues to produce humic acidand fulvic acid, which chelate with Ca and Fe in composite phosphate, thus releasing phosphate ions . (5) Phosphate-solubilizing microorganisms chelate Ca2+,thus allowing phosphate ions bound with Ca2+ to solubilize in the soil solution.

So, there are many mechanisms by which phosphorus-solubilizing microorganisms dissolve insoluble phosphorus. Although strain M2 secretes organic acids to dissolve insoluble phosphorus, the content of organic acids may not be high. In addition, strain M2 may also be able to secrete some alkaline substances to increase the pH of the solution. Therefore, this is not contradictory. In subsequent studies, we plan to use metabolomics to determine the composition of the strain M2 fermentation broth and explore the specific mechanism by which strain M2 increases the pH of the solution.

We have included these explanations in the discussion.

In solution culture, a change of 0.2 units in pH is very important. In particular, in the induction of phosphate mineralization by strain M2, a weakly alkaline pH can more effectively promote the formation of heavy metal phosphate precipitates.

Comments 8: Line 106: “white spots (containing Cd)” - Based on what data do the authors believe that the white spots contain cadmium?

Response 8: Thank you for pointing this out. Actually, we did perform energy spectrum identification for these white spots and found that they contained Cd. The energy spectrum diagram is now shown in Fig. S2.

After freeze-drying, the bacterial precipitates were observed by scanning electron microscopy coupled with energy dispersive X-ray (SEM-EDX) to determine their morphology. Under Cd stress, some cells of strain M2 exhibited a concave shape and many white spots (containing Cd) on the cell wall according to SEM‒EDS images (Fig. 1e and f, Fig. S2).

Comments 9: Line 108: How was the content of “free Cd” determined? Do the authors consider cadmium cations bound to organic acids to also be “free Cd”?

Response 9: Thank you for pointing this out. In our study, the cell wall of strain M2 was able to immobilize Cd. In addition, strain M2 could also induce phosphate precipitation of Cd, further reducing the content of available Cd in the solution. By centrifugation, the Cd immobilized by the strain and the Cd precipitated by phosphate could be separated, and the remaining Cd in the culture medium, which was the available Cd, could be determined by acid hydrolysis. To avoid ambiguity, we have modified ‘free Cd’ to ‘available Cd’. Cd bound to organic acids may further bind to bacterial cells or phosphate precipitates, therefore it is not an available form of Cd.

Comments 10: Lines 128-129: “other researchers have reached similar conclusions [29,30].” - this information should be moved to the Discussion section.

Response 10: Agree. We have done as suggested.

Comments 11: Line 145: Why do the authors interpret a decrease in enzyme activities (SOD, POD, and CAT) as “improved wheat resistance to Cd”.

Response 11: Thank you for pointing this out. We have made this correction as suggested. “...in wheat roots and leaves, indicating that strain M2 reduced the toxic effect of Cd on wheat.”

When plants are subjected to heavy metal stress, they adjust their metabolism and increase the activity of antioxidant enzymes to enhance their resistance to heavy metals. Strain M2 can reduce the absorption of Cd by wheat roots through adsorption and induced precipitation, alleviating the toxic effects of heavy metals on wheat. When the external stress suffered by wheat decreases, the activity of antioxidant enzymes also decreases.

In fact, the transcriptomic results also indicated that strain M2 increased the expression of genes related to Cd detoxification and Cd resistance in wheat roots. Therefore, strain M2 also enhanced the resistance of wheat to Cd. However, this conclusion was not appropriate to include in this paragraph about antioxidant enzymes.

Comments 12: Figure 3: Why is the one-way ANOVA done without dividing into “Root” and “Leaf” groups, as in Figures 2c and 2d?

Response 12: Thank you for pointing this out. We have made this correction as suggested.

Comments 13: Figures 4c and 4d: Provide the results of the statistical analysis of the results.

Response 13: Thank you for pointing this out. We have done as suggested.

Comments 14: Line 176: “A total of 227.24 Gb of clean data were obtained” - Explain what data you mean.

Response 14: The total of 227.24 Gb of clean data refers mainly to the raw data of wheat root genes (BioProject ID: PRJNA1133704). Total RNA was extracted from wheat roots, and mRNA was isolated from the total RNA for analysis of transcriptome information by A-T base pairing with polyA using magnetic beads with Oligo(dT). The second-generation high-throughput sequencing platform is designed to sequence short sequence fragments, and the enriched mRNA is a complete RNA sequence with an average length of several kb, so it needs to be interrupted randomly. By adding fragmentation buffer and choosing appropriate conditions, the mRNA could be randomly broken into small fragments of approximately 300 bp. Under the action of reverse transcriptase, a single strand cDNA was synthesized from the mRNA template by random primers, and then a stable double-strand structure was formed by double-strand synthesis.

Comments 15: Lines 192-194: “In addition, six comparison groups were established: M2+Cd vs. Cd, Cd vs. CK, M2+Cd vs. M2, M2 vs. Cd, M2+Cd vs. CK, and M2 vs. CK." - This proposal can be deleted since all possible pairs are indicated, and they are discussed further.

Response 15: Thank you for pointing this out. We have done as suggested.

Comments 16: Figure 5c: The authors should explain in more detail in the caption to the figure and in the text of the manuscript what the numbers indicated in the figure mean. In particular, explain what 726 (1.30%) means for "CK". This is 726 DEGs relative to which option? If these are 726 genes, the expression of which is represented only in the control variant, then how do they differ from the 254 (0.46%) DEGs for the three variants without control? Overall, the discussion of Figure 5c is very brief, leaving the interpretation of most of the values ​​in this figure unclear. Perhaps it would be clearer to show two figures instead of Figure 5c: a diagram for up-regulated genes and a diagram for down-regulated genes.

Response 16: Figure 5c shows Venn diagrams. Venn diagram analysis of the samples showed the number of genes that were commonly or specifically expressed between samples or groups. This study investigated the effect of strain M2 on the expression of wheat root-related genes under Cd stress. Figure 5c shows the analysis of genes shared and specifically expressed in wheat roots under the four treatments. GO enrichment analysis could be performed on the specifically expressed genes to understand their main biological functions. The results showed that there were 46,058 genes (85.77%) shared by all treatments. The number of unique genes in the CK treatment was 726 (1.3%), the number of unique genes in the Cd treatment was 1,957 (3.52%), the number of unique genes in the M2 treatment was 725 (1.3%), and the number of unique genes in the M2+Cd treatment was 523 (0.94%). In addition, there were 254 genes shared by the Cd, M2, and M2+Cd groups (0.46%), 563 genes shared by the CK, M2, and M2+Cd groups (1.01%), 1,400 genes shared by the CK, Cd, and M2+Cd groups (2.52%), and 656 genes shared by the CK, Cd, and M2 groups (1.18%).

The focus of the study was to screen out differentially expressed genes, and the screening of differentially expressed genes was conditional, mainly involving the comparison between two treatments. Figure 5d shows the number of differentially expressed genes (upregulated genes and a diagram for downregulated genes) between treatments and the specific functions.

We have revised Figure 5 as suggested.

Comments 17: Figure 7: The authors should improve the quality (contrast) of the drawing so that the inscriptions can be read.

Response 17: Thank you for pointing this out. We have revised the Figure 7 as suggested.

Comments 18: Section 2.9 contains a figure that does not have a link in the text or a caption.

Response 18: Thank you for pointing this out. We have made this correction as suggested.

Figure 8. Gene expression levels were determined by qRT–PCR.. CK: wheat roots not treated with Cd and not inoculated with strain M2; Cd: wheat roots treated with 1 mg L-1 Cd; M2: wheat roots inoculated with strain M2; and M2+Cd: wheat roots treated with 1 mg L-1 Cd and inoculated with strain M2. The data are presented as the means ± SD (n = 3). The bars indicated by different letters are significantly different (P < 0.05) according to one-way ANOVA.

Comments 19: Line 321: “tomato (Solanum tuberosum L).” Are the authors talking about tomatoes or Solanum tuberosum L.?

Response 19: Thank you for pointing this out. This is a writing error, tomato (Solanum lycopersicum. L) is correct. We have made this correction as suggested.

Comments 20: Line 328: Do the authors use the term “nutrients” to refer to cadmium?

Response 20: Agree. We have made this correction as suggested.

Comments 21: Line 340: Explain what is meant by “dissociated H+ can adsorb heavy metals.” How do hydrogen cations interact with metal cations???

Response 21: Thank you for pointing this out. This was a writing error. This section mainly discusses how the walls of plant root cells immobilize heavy metals. H+ has been revised to HCO3-.

Comments 22: Lines 358–359: Based on what data are conclusions 2 and 3 drawn?

Response 22: These conclusions mainly come from Figure 4.

In the Cd treatment, strong fluorescence signals were detected both on the root surface and in the root interior. The intensity of the fluorescence signals in the root tip region was greater than that in the mature region, while the intensity of the fluorescence signals on the root surface was greater than that in the root interior, indicating that Cd was enriched mainly on the root surface and in the root tip (Fig. 4a). In the M2+Cd treatment, fluorescence signals were detected only on the surface of wheat roots, and almost no fluorescence signals were detected in the root interior (Fig. 4b). Moreover, the Cd content on the wheat root surface in the Cd group was 0.54 mg kg-1, while strain M2 significantly (P<0.05) reduced (40.1%) the Cd content on the root surface (Fig. 4c). These findings indicated that strain M2 can inhibit Cd entry into the interior of wheat roots. In addition, the percentages of Cd distributed in the cell wall, organelle and soluble fractions of wheat roots in the Cd treatment were 33.6%, 41.5% and 24.9%, respectively (Fig. 4d). However, strain M2 significantly (P<0.05) reduced the percentage of Cd in the wheat root organelles and soluble fraction and increased the percentage of Cd in the wheat root cell walls (Fig. 4d), which further indicated that strain M2 had the ability to enhance the retention of Cd in the cell wall in wheat roots.

Comments 23: In general, the manuscript fragment from lines 332 to 359 lacks references to the work of other authors.

Response 23: Thank you for pointing this out. We agree with this comment. Therefore, We have added references in lines 332 to 359.

Comments 24: Lines 366-369: The proposal is not consistent with the results presented.

Response 24: Thank you for pointing this out. We have revised it as suggested.

In the M2+Cd vs. Cd and M2 vs. Cd comparisons, the number of downregulated DEGs in the roots was greater than the number of upregulated DEGs (Fig. 5d), indicating that strain M2 reduced the toxicity of Cd to wheat roots through metabolism or induced the expression of genes in wheat roots, thereby eliminating the need for the overexpression of certain genes.

Comments 25: Line 376: What evidence shows that strain M2 increased the expression of glutathione reductase (GR)?

Response 25: Thank you for pointing this out. This was a writing error. We have removed the relevant description.

Comments 26: Line 444: What is GSG?

Response 26: Thank you for pointing this out. This was a writing error. Glutathione (GSH) was right. We have made this correction as suggested.

Comments 27: Lines 452-453: To what volume of LB medium was 0.3 mL of the seed solution added?

Response 27: This was the step to activate strain M2. The seed solution (0.3 mL) was inoculated into 10 mL Luria–Bertani liquid medium (5.0 g L-1 beef extract, 10.0 g L-1 peptone, 5.0 g L-1 NaCl; pH 7.0) and incubated at 30°C for 24 h.

Comments 28: Line 464: What cadmium compound was used in the experiments?

Response 28: Cd(NO3)2 was used; we have added this information in the text.

Comments 29: Line 479: Write the scientific name of common wheat (Triticum aestivum L.).

Response 29: Thank you for pointing this out. We have made this correction as suggested. Several seeds of the wheat (Triticum aestivum L., JiMai-22) were selected.

Comments 30: Lines 481-484: Explain what “the upper layer of a pot” and “the lower layer of a pot” mean.

Response 30: This is a commonly used device for hydroponic experiments. This pot has two parts. The upper part is mainly used to fix wheat, and there are many small holes in the middle. The wheat roots can enter the lower layer of culture solution through the holes. A photograph of this device is shown below.

Fig. 1 Hydroponic apparatus diagram

Round 2

Reviewer 2 Report

Comments and Suggestions for Authors

I agree with most of the authors' responses. However, I have two comments for the revised manuscript:

1. Lines 36–37: I disagree with the statement "Wheat (Triticum aestivum L.) is the main food crop in China and is widely cultivated [3]." (See also the review of version 1). What meaning do the authors put into "the main food crop"? Reference [3] does not support this information. I suggest the authors delete this sentence or replace it with some more general information with an appropriate reference.

2. In response to comment 3, the authors indicate, "we have deposited strain M2 in the China Center for Type Culture Collection (CCTCC). Due to the time required for the preservation of bacterial strains, when we receive the proof of preservation, we added this information to section 4.1." This information about the strain should be submitted before accepting the manuscript.

Author Response

Comments 1: Lines 36–37: I disagree with the statement "Wheat (Triticum aestivum L.) is the main food crop in China and is widely cultivated [3]." (See also the review of version 1). What meaning do the authors put into "the main food crop"? Reference [3] does not support this information. I suggest the authors delete this sentence or replace it with some more general information with an appropriate reference.

Response 1: Thank you for pointing this out. We agree with this comment. Therefore, we have revised it as suggested.

In recent years, the continuous exposure of the "cadmium wheat" incident has caused widespread concern among the whole society about the issue of excessive Cd in wheat [3,4].

Comments 2: In response to comment 3, the authors indicate, "we have deposited strain M2 in the China Center for Type Culture Collection (CCTCC). Due to the time required for the preservation of bacterial strains, when we receive the proof of preservation, we added this information to section 4.1." This information about the strain should be submitted before accepting the manuscript.

Response 2: Thank you for pointing this out. We agree with this comment. Therefore, we have added added this information to section 4.1. The phosphorus-solubilizing bacterium M2 (CCTCC M 2024654), which was isolated from heavy metal-contaminated radish rhizosphere soil, can secrete organic acids to dissolve insoluble phosphorus [13]